# Non-pharmacological therapies for postviral syndromes, including Long COVID: a systematic review and meta-analysis protocol

Joht Singh Chandan [1,2] Kirsty Brown [1] Nikita Simms-Williams [1]
Jenny Camaradou,[3] Nasir Bashir,[4] Dominic Heining,[5]
Olalekan Lee Aiyegbusi [1,6,7,8,9] Grace Turner [1,6]
Samantha Cruz Rivera [1,6,8] Richard Hotham,[1] Krishnarajah Nirantharakumar,[1,2]
Manoj Sivan [10] Kamlesh Khunti [11] Devan Raindi,[4] Steven Marwaha,[12]
Sarah E Hughes [1,6,9] Christel McMullan [1,6] Melanie Calvert [1,2,6,7,8,9,13]
Shamil Haroon [1]

JSC and KB are joint first authors.

MC and SH are joint senior authors.

For numbered affiliations see end of article.

**Correspondence to**
Dr Joht Singh Chandan;
j.s.chandan.1@bham.ac.uk

## ABSTRACT

**Introduction** Postviral syndromes (PVS) describe the sustained presence of symptoms following an acute viral infection, for months or even years. Exposure to the SARS-CoV-2 virus and subsequent development of COVID-19 has shown to have similar effects with individuals continuing to exhibit symptoms for greater than 12 weeks. The sustained presence of symptoms is variably referred to as 'post COVID-19 syndrome', 'post-COVID condition' or more commonly 'Long COVID'. Knowledge of the long-term health impacts and treatments for Long COVID are evolving. To minimise overlap with existing work in the field exploring treatments of Long COVID, we have only chosen to focus on non-pharmacological treatments.
**Aims** This review aims to summarise the effectiveness of non-pharmacological treatments for PVS, including Long COVID. A secondary aim is to summarise the symptoms and health impacts associated with PVS in individuals recruited to treatment studies.
**Methods and analysis** Primary electronic searches will be performed in bibliographic databases including: Embase, MEDLINE, PyscINFO, CINAHL and MedRxiv from 1 January 2001 to 29 October 2021. At least two independent reviewers will screen each study for inclusion and data will be extracted from all eligible studies onto a data extraction form. The quality of all included studies will be assessed using Cochrane risk of bias tools and the Newcastle-Ottawa grading system. Non-pharmacological treatments for PVS and Long COVID will be narratively summarised and effect estimates will be pooled using random effects meta-analysis where there is sufficient methodological homogeneity. The symptoms and health impacts reported in the included studies on non-pharmacological interventions will be extracted and narratively reported.
**Ethics and dissemination** This systematic review does not require ethical approval. The findings from this study will be submitted for peer-reviewed publication, shared at conference presentations and disseminated to both clinical and patient groups.

### Strengths and limitations of this study

► This review will provide the first comprehensive and systematic summary of non-pharmacological treatments for postviral syndromes.
► The review proposal has been strengthened through its codevelopment with individuals with lived experience of Long COVID.
► A limitation of the review relates to the heterogeneity in the definition of post-viral syndromes as well as Long COVID, adding complexity to the synthesis of available evidence across the included syndromes.
► There is likely to be significant methodological heterogeneity in clinical trials of non-pharmacological interventions for postviral syndromes, which may limit our ability to pool treatment effects.

**PROSPERO registration number** The review will adhere to this protocol which has also been registered with PROSPERO (CRD42021282074).

## BACKGROUND

Globally, there have been over 480 million cases of COVID-19 with over 6 million associated deaths.[1] The pandemic has triggered a concerted global effort to develop and deliver safe and efficacious vaccinations at record speed, which have now significantly reduced morbidity, mortality and disease transmission associated with SARS CoV-2.[2–4]

Despite such positivity, the long-term sequalae of even mild infections of COVID-19 appear to carry a substantive burden on some patients, their families and health services. Early evidence suggested that the average recovery time following an acute COVID-19 infection is 2–3 weeks, depending

on symptom severity.[5][6] However, as the pandemic has progressed it became apparent that not all patients immediately recovered from the initial infection and between 2.3% and 37% develop symptoms lasting greater than 12 weeks, some of which may be relapsing and remitting.[7][8] However, prevalence estimates in these longer-term symptoms vary because of differences in disease definition (including differences in symptom severity and included symptoms), population surveyed and study design.[8–13]

There is considerable heterogeneity in the clinical definition of ongoing symptoms following initial exposure to SARS CoV-2. The National Institute for Health and Care Excellence define symptoms persisting beyond 4 weeks as either 'ongoing symptomatic COVID-19' (symptoms lasting between 4 and 12 weeks) or 'Post COVID-19 syndrome' (symptoms lasting beyond 12 weeks).[14] Globally other terms such as 'post-COVID condition' or 'long haulers' are used. However, in the UK, the more commonly adopted and preferred term among healthcare professionals and patient groups for ongoing symptoms following initial viral exposure to SARS CoV-2 is 'Long COVID,' the term adopted for this review.

The prevalence of persistent signs and symptoms, the remitting, relapsing and interchangeable nature of these symptoms, the pathophysiology and clinical course of Long COVID, and the lived experiences of patients, is yet to be fully described. However, many clinical sequalae including, but not limited to, fatigue, respiratory, neurological, psychological and gastroenterological complications that present in patients with Long COVID also feature in a similar pattern following other acute viral infections.[13][15]

An improved understanding of the sequalae and particularly the management of existing post-viral syndromes secondary to acute infections may provide further insight and guidance as to the optimal management of patients with Long COVID. Despite the novelty of the condition, it is clear that optimal management involves a multidisciplinary approach and rehabilitation, with patient involvement being a central component.[16][17]

Postviral syndromes may occur secondary to exposure to a number of acute viral infections, including but not limited to Coronaviridae, Filoviridae, Herpesviridae, Orthomyxoviridae, Picornaviridae, Parvoviridae, Togaviridae and Flaviviridae, as defined by the International Committee on Taxonomy of Viruses.[18–26] Like Long COVID, the signs and symptoms comprising these postviral syndromes may include fatigue, musculoskeletal pain (eg, joint pain) and mood (eg, feeling depressed) and neurocognitive disturbances (eg, difficulties with concentration and sleep).[15][27] A particularly researched area has been the development of postinfectious fatigue following Epstein-Barr virus (EBV) exposure.[28] As we have demonstrated in our recently published review on the symptoms of Long COVID, fatigue is one of the most commonly reported symptoms.[29] Notably, over time interventions have been developed to support patients with Epstein-Barr postinfectious fatigue, such as the

combination of cognitive–behavioural therapy with music therapy which has shown some promise.[30] This example highlights the similarities that may exist between postviral syndromes and the need to summarise evidence on their treatments, which may be relevant to those experiencing Long COVID.

Although the pathophysiology of postviral syndromes is poorly understood, it has been hypothesised that their development is mediated by altered cytokine responses, neuroinflammation, mitochondrial dysfunction and microbiome dysbiosis.[31–35] A culmination of the pathophysiological changes and phenotypes following acute viral infections, including COVID-19, have been likened to other poorly understood conditions such as myalgic encephalomyelitis (ME)/chronic fatigue syndrome (CFS).[36] Despite the limited evidence and clinical consensus, guidelines for the management of Long COVID have been developed and trials are underway exploring the impact of pharmaceutical, dietary and lifestyle interventions.[14][37]

Considering the significant individual and societal burden arising from the COVID-19 pandemic there is an urgent need to identify whether the existing knowledge base on similar postviral syndromes can be applied to support the clinical management of patients with Long COVID. A comprehensive synthesis of the evidence on postviral syndromes, including on their therapies, is needed to better support those with Long COVID as well as those with previous post-viral syndromes. To minimise overlap with existing work in the field exploring pharmacological treatments for Long COVID and on the basis of the advice of our expert advisory group, we have chosen to focus on non-pharmacological treatments.

## OBJECTIVES

This systematic review will summarise the evidence on non-pharmacological treatments for post-viral syndromes, including Long COVID. The findings will be used to make recommendations on non-pharmacological treatments for Long COVID as well as previous postviral syndromes.[38]

This systematic review will be conducted and reported in accordance with the Preferred Reporting Items for Systematic Reviews and Meta-Analyses (PRISMA) guidelines, and this protocol has been written in accordance with the PRISMA-Protocols checklist (online supplemental file 1).[39][40]

## METHODS
### Eligibility criteria

We will include primary research studies. For non-pharmacological treatments effects, we will consider randomised and non-randomised controlled trial evidence evaluating interventions for patients with postviral syndromes and will additionally include observational studies (cohort, case cross-over and case–control study designs) for studies focusing on Long COVID.

There will be no restrictions on setting (ie, community or hospital based). Studies published in the previous 20 years will be included (study period defined as from 1 January 2001 to 29 October 2021), as this would encompass research on the first SARS and MERS outbreak which are the two pandemic viruses most related to SARS-CoV-2. No language restrictions will be applied. Peer-reviewed published studies, preprint articles and other grey literature will also be included in this review.

We will include studies with a population consisting of adults and/or children with postviral syndromes. Studies investigating postviral syndromes following acute infections from the following families (grouped by the International Committee on Taxonomy of Viruses)[18] will be included in this review.

Viruses known to cause symptomatic acute viral infections will be included. This is based on an assessment of clinically relevant viruses extracted from 'Mandell, Douglas, and Bennett's Principles and Practice of Infectious Disease, ninth Edition.'[41] Included in brackets are the relevant clinical infections used to inform search terms:

Coronaviridae (Middle East Respiratory Syndrome-Coronavirus, Severe Acute Respiratory Syndrome-Coronavirus, and other coronaviruses), Filoviridae (Ebola, Marburg), Herpesviridae (Epstein-Barr Virus, Cytomegalovirus, Varicella Zoster Virus, Other Human Herpes Viruses, Herpes Simplex Virus), Orthomyxoviridae (Influenza, A, B, C and D), Picornaviridae (Coxsackie, Hepatitis A, Enterovirus including Rhinovirus), Parvoviridae (Parvovirus), Togaviridae (Ross River virus, Chikungunya, other Semliki Forest Complex Viruses), Flaviviridae (West Nile River Virus, Dengue, Yellow Fever, Zika), Reoviridiae (Rotavirus), Paramyxoviridae (Mumps, Measles, Parainfluenza, Respiratory Syncytial Virus, Human Metapneumovirus) Adenoviridiae (Adenoviruses), Caliciviridae (Norovirus, Sapovirus), Astroviridae (Astroviruses). Additionally, when examining treatment options for Long COVID we will also include studies relating to SARS-CoV-2.

Extremely rare viral infections will not be included, as there is unlikely to be published research relevant to this review. Predominantly asymptomatic viruses, viruses identified in the last 3 years (except for SARS-CoV-2), and viruses predominantly existing as zoonoses will all be excluded, in addition to studies on chronic viral infections including Hepatitis B, Hepatitis C and HIV postviral syndromes. Studies which include demonstrable reactivation of latent infection (eg, EBV, CMV with detectable viral load) will be excluded. We will exclude in vitro and animal studies. Studies that principally focus on ME and CFS without reference to pre-specified viruses will also be excluded.

To the authors' knowledge, there is no internationally agreed definition of post-viral syndromes and as such there is likely to be heterogeneity in the temporal description of syndrome onset following the initial viral exposure. Where temporal criteria are included, we will aim to include those with symptoms lasting beyond 12 weeks (in line with the minimum timeframe for Long COVID used by the WHO).[42] However we will also include publications which provide no firm timeframe but indicate an aspect of chronicity or symptoms persisting for a prolonged period. Additionally, like Long COVID, patients with post-viral syndromes can experience a wide range of symptoms ranging from respiratory complications to fatigue or musculoskeletal pain.

The outcomes of the study will be changes in symptoms, exercise capacity, quality of life (including changes in mental health and well-being) and work capability. Although these are numerous and heterogeneous in nature, some examples of non-pharmacological treatments which may be used to support patients include pacing, cognitive behavioural therapy and pulmonary rehabilitation. Additionally, where described in the included studies, we will also extract information on the symptoms and health impacts reported by participants in the included studies.

In summary, the population will consist of either people with a post-viral syndrome who contributed to a randomised controlled trial or those with Long COVID who either contributed to an observational study or a randomised controlled trial. We will include studies that have assessed the effectiveness of non-pharmacological interventions designed to improve symptoms of postviral syndromes against standard care, an alternative non-pharmacological therapy, or a placebo. The outcomes will be changes in symptoms, exercise capacity, quality of life (including changes in mental health and well-being), and work capability.

## Information sources

Primary electronic searches will be performed in Embase, MEDLINE, PyscINFO, CINAHL, COVID-NMA and preprint servers (MedRxiv up to the first 500 references) from 1 January 2001 to 29 October 2021. The reference lists of all eligible articles will also be searched to identify any additional eligible studies which were not identified during the original search. Secondary searches will be conducted for grey literature on databases such as Google Scholar (limited to the first 500 references). Abstracts, conference and symposia proceedings from relevant organisations relating to virology will be identified and examined.

## Search methods for identification of studies

The search strategy broadly includes terms such as: "MERS-COV", "SARS-COV", "Coronavirus", "Ebola", "Epstein-Barr Virus", "EBV", "Cytomegalovirus", "CMV", "Herpes", "Influenza", "Swine flu", "Coxsackie", "Parvovirus", "Ross River virus", "Chikungunya", "West Nile River virus", "long COVID", "post-acute-COVID", "Dengue", "treatment", "management", "Post-viral", "quality of life", "outcomes". A more extensive list of search terms is attached in online supplemental file 2.

Studies will not be excluded on language. Where studies are not published in English then they will be translated using Google Translate or a by a member of staff at the Institute of Applied Health Research.

## Study records

### Data management, selection and data collection processes

The study selection procedure will consist of four steps documented using a PRISMA flow diagram with reviewer decisions recorded at each stage:

1. The search results will be downloaded onto an excel file which will be imported onto the online review software COVIDENCE where the results will be deduplicated.[43] Titles with abstracts will be initially screened by at least two reviewers using predefined screening criteria based on whether studies (1) included patients with a diagnosis of a post-viral syndrome and (2) included information about non-pharmacological treatments.
2. Following title and abstract screening by two reviewers, any disagreements will be resolved through discussion. Where disagreements remain following discussion, these will be reconciled involving an additional reviewer.
3. Full texts of the selected articles will be obtained, and exclusion criteria applied by at least two independent reviewers.
4. As in step 2, selection of included articles will be discussed until a consensus is reached between the reviewers, with an additional reviewer providing support to resolve disagreements. The conclusion of step 3 will provide the final list of studies eligible for inclusion in the review.

The data from all eligible studies will be extracted onto an extraction form by at least one reviewer, with a quarter of extracted studies checked by a second reviewer. In the event where the full text is not available, we will contact the study's corresponding author for access either via their email address provided for the article or ResearchGate.

### Data items

Data will be extracted for all eligible studies including first author, year of publication, study design, study setting (community or hospital), country, participant characteristics (age, sex, ethnicity), name of virus and outcome measures. A draft of the data extraction tables can be seen in online supplemental file 3.

### Outcomes and prioritisation

Data will be sought for the type of non-pharmacological treatment/intervention. The outcomes will be changes in symptoms, exercise capacity, quality of life (including changes in mental health and well-being) and work capability. Data on the symptoms, health impacts and complications experienced by patients at study entry included in the studies will also be extracted where available.

### Risk of bias in individual studies

To assess the risk of bias in individual studies we will use the appropriate tool for each type of study. Cochrane risk of bias tools will be used for randomised (RoB 2) and non-randomised (ROBINS-I) controlled trials, and the Newcastle Ottawa tool will be used for observational studies.[44–47] The risk of bias will be examined by two reviewers and reported using the respective tools.

### Data synthesis

The review findings will present the non-pharmacological treatments for previous postviral syndromes and Long COVID.

The review will consist of a narrative and quantitative synthesis of included studies if appropriate. The characteristics (study design, population, setting, viruses, etc) and findings of included studies will first be narratively described and summarised in tables. The outcomes (treatment effects (mean differences or measures of relative risk)) will be presented on forest plots to assess the heterogeneity of study findings. The methodological heterogeneity of studies will be qualitatively evaluated and then assessed using the $I^2$ statistic. We will assess clinical heterogeneity by examining the types of participants, interventions and outcomes in each study. We anticipate considerable heterogeneity, particularly in exposure definition (post-viral syndromes or Long COVID), due to differing temporal definitions, viral types and nature of symptoms, and thus where meta-analysis is considered appropriate will undertake a random effects meta-analysis rather than a fixed effect model.

Where outcomes are measured using a continuous scale, these will be pooled through calculation of the weighted mean difference (with SD). Where outcomes are assessed using different continuous measures across the studies, these will be pooled using Hedges' $g$ standardised mean difference. Where outcomes are assessed using a dichotomous measure, these will be pooled through calculation of ORs.

In line with Cochrane guidance for systematic reviews on dealing with missing data, when appropriate we will not impute missing data but will omit the data from the respective analysis being conducted.[48]

### Additional analyses

Where there is sufficient data, we will conduct a subgroup analysis comparing hospitalised to non-hospitalised patients and observing any differences in outcomes. We will also where possible undertake a meta-regression model to assess the impact of age and sex on symptoms and treatment effects. Additionally, as a sensitivity analysis we will repeat the primary analysis using high quality studies only.

If at least 10 studies are included in the meta-analysis with no significant evidence of heterogeneity a funnel plot will be used to explore the existence of publication bias by visual inspection. We will also undertake an objective assessment of publication bias using Egger's regression test. Asymmetry of the funnel plot and/or statistical significance of Egger's regression test ($p<0.05$) will be indicative of publication bias.

## Confidence in cumulative evidence

At least two independent reviewers will assess the strength of the body of evidence (according to the following categories: high, moderate, low and very low) for each individual outcome using the five Grades of Recommendations, Assessment, Development and Evaluation (GRADE) considerations for downgrading the certainty of evidence (risk of bias, impression, inconsistency, indirectness and publication bias) and the three GRADE considerations for upgrading the certainty of evidence (large magnitude of effect, dose-response gradient and residual confounding decreasing the overall effect).[49]

## Patient and public involvement

Patients and the public were involved in the development of the research question and design of the review. This was undertaken with the support of a working patient and public involvement and engagement (PPIE) group. We held an initial meeting with members of the PPIE group. Members of the research team gave a presentation which highlighted the similarity between postviral syndromes and Long COVID, the rationale for this review as well as our preliminary ideas. There were four PPIE group members in attendance—two white men and two women (one black and the other South-East Asian). Following the presentation, there was a discussion of the review objectives. The PPIE group members were asked for their thoughts on the review objectives and its scope. While they confirmed that the objectives were appropriate and comprehensive, they stressed the impact of Long COVID on employment and that the loss of income was very important to patients. Therefore, a key goal of this work should be to identify potential non-pharmacological interventions that will facilitate the rehabilitation of patients with long COVID so that they can be fit to return to work. Based on their feedback, we will ensure that this review will report interventions designed to improve the physical function of patients which will facilitate their return to work. Patients and members of the public will not be involved in the study selection or extraction phases but will be involved with the evidence synthesis stage. At this subsequent stage their involvement will be documented according to the GRIPP-2 checklist.[50] The final review, and a lay summary will be disseminated via publication and conference presentations which will be supported with the PPIE group.

## ETHICS AND DISSEMINATION

As the primary focus of the review will be to undertake a secondary analysis of published data, no ethical approval is formally required. The findings from the review will be submitted for peer-reviewed publication, shared at conference presentations, and disseminated to both clinical and patient groups.

**Author affiliations**
[1]Institute of Applied Health Research, University of Birmingham, Birmingham, UK
[2]Health Data Research UK, Birmingham, UK
[3]COVID END Evidence Network, UK, UK
[4]School of Dentistry, University of Birmingham, Birmingham, UK
[5]University Hospitals Birmingham NHS Foundation Trust, Birmingham, UK
[6]Centre for Patient-Reported Outcomes Research, University of Birmingham, Birmingham, UK
[7]NIHR Birmingham Biomedical Research Centre, University Hospitals Birmingham NHS Foundation Trust, Birmingham, UK
[8]Birmingham Health Partners Centre for Regulatory Science and Innovation, University of Birmingham, Birmingham, UK
[9]National Institute for Health Research (NIHR) Applied Research Centre West Midlands, University of Birmingham, Birmingham, UK
[10]University of Leeds, Leeds, UK
[11]Department of Health Sciences, University of Leicester, Leicester, UK
[12]Institute for Mental Health, University of Birmingham, Birmingham, UK
[13]Centre for Patient Reported Outcomes Research and Institute of Applied Health Research, University of Birmingham, Birmingham, UK

**Contributors** The research questions were conceived by JSC, KN, MC and SH. The first draft of the protocol was done by KB, NS-W, JSC and SH. Further reviews and revisions to the protocol were made by all remaining co-authors (CM, JC, NB, DH, OLA, GT, KN, MS, KK, DR, SCR, RH, SM, SEH). The final copy of the manuscript was approved by all authors.

**Funding** This work is independent research jointly funded by the National Institute for Health Research (NIHR) and UK Research and Innovation (UKRI) (Therapies for Long COVID in non-hospitalised individuals: From symptoms, patient reported outcomes and immunology to targeted therapies (The TLC Study), COV-LT-0013).

**Disclaimer** The views expressed in this publication are those of the author(s) and not necessarily those of the National Institute for Health Research or the Department of Health and Social Care.

**Competing interests** None declared.

**Patient and public involvement** Patients and/or the public were involved in the design, or conduct, or reporting, or dissemination plans of this research. Refer to the Methods section for further details.

**Patient consent for publication** Not applicable.

**Provenance and peer review** Not commissioned; externally peer reviewed.

**ORCID iDs**
Joht Singh Chandan http://orcid.org/0000-0002-9561-5141
Kirsty Brown http://orcid.org/0000-0001-9403-1593
Nikita Simms-Williams http://orcid.org/0000-0002-4926-1995
Olalekan Lee Aiyegbusi http://orcid.org/0000-0001-9122-8251
Grace Turner http://orcid.org/0000-0002-9783-9413
Samantha Cruz Rivera http://orcid.org/0000-0002-1566-6804
Manoj Sivan http://orcid.org/0000-0002-0334-2968
Kamlesh Khunti http://orcid.org/0000-0003-2343-7099
Sarah E Hughes http://orcid.org/0000-0001-5656-1198
Christel McMullan http://orcid.org/0000-0002-0878-1513
Melanie Calvert http://orcid.org/0000-0002-1856-837X

## Open access

Shamil Haroon http://orcid.org/0000-0002-0096-1413

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
