## [Reviewer comments · BMJ Open]

ARTICLE DETAILS

TITLE (PROVISIONAL)	Non-pharmacological therapies for post-viral syndromes, including Long COVID: A systematic review and meta-analysis protocol
AUTHORS	Chandan, Joht; Brown, Kirsty; Simms-Williams, Nikita; Camaradou, Jenny; Bashir, Nasir; Heining, Dominic; Aiyegbusi, Olalekan Lee; Turner, Grace; Cruz Rivera, Samantha; Hotham, Richard; Nirantharakumar, Krishnarajah; Sivan, Manoj; Khunti, Kamlesh; Raindi, Devan; Marwaha, Steven; Hughes, Sarah; McMullan, Christel; Calvert, Melanie; Haroon, Shamil

VERSION 1 – REVIEW

REVIEWER	Stephenson, Terence University College London, Institute of Child Health I know some of the authors but have not published with them although I may do in the future. I have a research grant to study long covid. I am a co-author on a systematic review with meta-analysis of persisting symptoms following SARS-COV-2 which has been submitted and is under review.
REVIEW RETURNED	27-Oct-2021

GENERAL COMMENTS	A very important topic. The authors rightly remind us that rather than re-invent the wheel we should certainly first conduct a systematic literature search and evidence synthesis to discover and summarise what is already known. In the case of Long Covid, this will involve extrapolating from other post-viral syndromes which occurred before the pandemic and therefore I believe this is a very important systematic review and meta-analysis which will inform practice and help decide future management. Frankly, it is surprising that the topic of this systematic review has not already been conducted and published so congratulations to the authors for pursuing this. The authors recognise that a limitation of the review relates to the heterogeneity in the definition of post-viral syndromes. This needs to be discussed in more detail. This review aims to summarise the symptoms, health impacts and treatments for existing post-viral syndromes, to derive evidence relevant to the management of Long COVID. In the present pandemic, it is all too easy to presume this is a systematic review of long Covid and the fact that it is actually about re-visiting what is already known about previous post-viral syndromes could be highlighted more prominently by the use of bold text, underlining, repetition or some other device. Readers scanning an abstract
---

	quickly may otherwise miss this important point. It is a very ambitious protocol and I wonder if the protocol should specify that the final published results may split the systematic review and meta-analysis into reporting:  1. persisting symptoms of post-viral syndromes separately from 2. health impacts of post-viral syndromes (clinical complications and impacts on quality of life and work capability) 3. treatment of post-viral syndromes. In relation to the latter, "the review will summarise evidence on non-pharmacological treatments for previous post-viral syndromes and Long COVID. The findings will be used to make recommendations on non-pharmacological treatments for Long COVID as well as previous post-viral syndromes." This is confusing because, as already noted above, the authors claim this is a systematic review of pre-existing knowledge about known post-viral syndromes. Now it looks as if they are going to review both known post-viral syndromes and emerging evidence on long-Covid? "We have focused on non-pharmacological treatments as pharmacological therapies are currently already reviewed by the UK COVID-19 Therapeutics Panel (UK-CTAP)." This makes me think that both the title and the abstract are unbalanced and inaccurate since they both imply that all treatments, without qualification, will be covered in the review whereas it is only in the body of the text that it becomes clear that the review will confine itself to non-pharmacological treatments. "Patients and the public were involved in the development of the research question and design of the review. This was undertaken with the support of a working patient and public involvement (PPI) group." This is rather telegraphic. How were patients and the public involved, what was their contribution, what changes did they recommend that were accepted or declined, what was the composition of the working PPI group and what is the evidence of diversity in that group?
--	---

REVIEWER	Khanpour Ardestani, Samaneh University of Alberta Faculty of Medicine and Dentistry, Pediatrics
REVIEW RETURNED	10-Nov-2021

GENERAL COMMENTS	The presented paper proposes a protocol for conducting a systematic review on post-viral syndromes. The implications will be relevant for the management of similar syndromes and long-COVID. The objective of the study is extensively broad and tries to cover all aspects of these conditions. However, as briefly mentioned by authors, the heterogeneity in definitions of these syndromes and the outcome measures used by different studies will limit the generalizability of the evidence. Some issues should be addressed for proper and transparent understanding of the review procedures. Please find some suggestions below: Title: Please identify in the title that this study is a systematic review protocol Abstract-methods: January 2001 to inception: Do you mean January 2001 to present? Please clearly describe your inclusion criteria based on PICO-S Background: Page 8, line 6: Please update the numbers and indicate the date as the cases are still evolving.
---

	Page 8, paragraph 5: You may want to explain more about epidemiological features of post-viral syndromes to build a stronger support for your background. Objectives: Please indicate your PROSPERO registration number Methods: Eligibility criteria-I see that this review intends to describe what is out there in terms of post-viral syndromes (symptoms, interventions, outcomes). However, for this review to be more transparent, I suggest providing a potential list of what you will consider included with regard to these variables (in a table or text). Especially this would be helpful for non-pharmacological treatments. Outcomes and prioritisation: As mentioned above, providing a potential list of what this review will consider as outcomes in detail will help.
--	---

VERSION 1 – AUTHOR RESPONSE

Reviewer 1:

Comment 1:

A very important topic. The authors rightly remind us that rather than re-invent the wheel we should certainly first conduct a systematic literature search and evidence synthesis to discover and summarise what is already known. In the case of Long Covid, this will involve extrapolating from other post-viral syndromes which occurred before the pandemic and therefore I believe this is a very important systematic review and meta-analysis which will inform practice and help decide future management. Frankly, it is surprising that the topic of this systematic review has not already been conducted and published so congratulations to the authors for pursuing this.

The authors recognise that a limitation of the review relates to the heterogeneity in the definition of post-viral syndromes. This needs to be discussed in more detail.

Response 1:

We thank Prof Stephenson for his kind comments about the review and its purpose. We agree with this point and have since refined the scope of the review. Now we have clarified in the manuscript we will be undertaking a single review focusing on non-pharmacological treatments of post-viral syndromes and Long COVID. This is because there have since been reviews published on symptoms and health impacts of some of the most closely linked coronaviruses (including Long COVID) with more due to be published in the coming months to our knowledge as part of the nationally funded Long COVID work.^{1,2} However, we will still extract data on symptoms and health impacts reported in included studies and include this within one review.

Comment 2:

This review aims to summarise the symptoms, health impacts and treatments for existing post-viral syndromes, to derive evidence relevant to the management of Long COVID. In the present pandemic, it is all too easy to presume this is a systematic review of long Covid and the fact that it is actually about re-visiting what is already known about previous post-viral syndromes could be highlighted more prominently by the use of bold text, underlining, repetition or some other device. Readers scanning an abstract quickly may otherwise miss this important point.

Response 2:

We thank the reviewer for highlighting this important point. We have attempted to create a paragraph break (if deemed suitable by the editorial team) entitled “aims:” to highlight the key aim of the study more prominently in the abstract as well as to demonstrate to readers that we will also be summarising the emerging evidence on non-pharmacological treatments for Long COVID.

Aims: This review aims to summarise the effectiveness of non-pharmacological treatments for post-viral syndromes, including Long COVID. A secondary aim is to summarise the symptoms and health impacts associated with post-viral syndromes in individuals recruited to treatment studies.

Comment 3:

It is a very ambitious protocol and I wonder if the protocol should specify that the final published results may split the systematic review and meta-analysis into reporting:

1. persisting symptoms of post-viral syndromes separately from
2. health impacts of post-viral syndromes (clinical complications and impacts on quality of life and work capability)
3. treatment of post-viral syndromes.

In relation to the latter, "the review will summarise evidence on non-pharmacological treatments for previous post-viral syndromes and Long COVID. The findings will be used to make recommendations on non-pharmacological treatments for Long COVID as well as previous post-viral syndromes." This is confusing because, as already noted above, the authors claim this is a systematic review of pre-existing knowledge about known post-viral syndromes. Now it looks as if they are going to review both known post-viral syndromes and emerging evidence on long-Covid?

Response 3:

We thank the reviewer for their comments on this point. Considering emerging evidence highlighted above, we have now clarified in the text that we will produce a single report examining non-pharmacological treatments of post-viral syndromes including Long COVID.

We are grateful for the reviewer's comment as we appreciate the submitted version of the manuscript was not clear in differentiating that we are also looking at the emerging evidence on Long COVID management in addition to post-viral syndromes. Therefore, we have made numerous edits throughout the manuscript to more clearly reflect this.

Comment 4:

"We have focused on non-pharmacological treatments as pharmacological therapies are currently already reviewed by the UK COVID-19 Therapeutics Panel (UK-CTAP)." This makes me think that both the title and the abstract are unbalanced and inaccurate since they both imply that all treatments, without qualification, will be covered in the review whereas it is only in the body of the text that it becomes clear that the review will confine itself to non-pharmacological treatments.

Response 4:

We thank the reviewer for this point and have now amended the abstract and title to include reference to only non-pharmacological treatments.

Comment 5:

"Patients and the public were involved in the development of the research question and design of the review. This was undertaken with the support of a working patient and public involvement (PPI) group." This is rather telegraphic. How were patients and the public involved, what was their contribution, what changes did they recommend that were accepted or declined, what was the composition of the working PPI group and what is the evidence of diversity in that group?

Response 5:

We thank the reviewer for the comment and have added further information in the manuscript to highlight the role of the PPI group and partners:

Patients and the public were involved in the development of the research question and design of the review. This was undertaken with the support of a working patient and public involvement and engagement (PPIE) group. We held an initial meeting with members of the PPIE group. Members of the research team gave a presentation which highlighted the similarity between post-viral syndromes and Long COVID, the rationale for this review as well as our preliminary ideas. There were four PPIE group members in attendance – two White men and two women (one Black and the other South-East Asian). Following the presentation, there was a discussion of the review objectives. The PPIE group

members were asked for their thoughts on the review objectives and its scope. While they confirmed that the objectives were appropriate and comprehensive, they stressed the impact of long COVID on employment and that the loss of income was very important to patients. Therefore, a key goal of this work should be to identify potential non-pharmacological interventions that will facilitate the rehabilitation of patients with long COVID so that they can be fit to return to work. Based on their feedback, we will ensure that this review will report interventions designed to improve the physical function of patients which will facilitate their return to work. Patients and members of the public will not be involved in the study selection or extraction phases but will be involved with the evidence synthesis stage. At this subsequent stage their involvement will be documented according to the GRIPP-2 checklist.³ The final review, and a lay summary will be disseminated via publication and conference presentations which will be supported with the PPIE group.

Reviewer 2:

Comment 1:

The presented paper proposes a protocol for conducting a systematic review on post-viral syndromes. The implications will be relevant for the management of similar syndromes and long-COVID. The objective of the study is extensively broad and tries to cover all aspects of these conditions. However, as briefly mentioned by authors, the heterogeneity in definitions of these syndromes and the outcome measures used by different studies will limit the generalizability of the evidence. Some issues should be addressed for proper and transparent understanding of the review procedures. Please find some suggestions below:

Title: Please identify in the title that this study is a systematic review protocol

Response 1:

We thank Dr Ardestani for their kind comments on the protocol. We have now decided to do a single review focusing on non-pharmacological interventions for post-viral syndromes and Long COVID and have reflected the title accordingly.

Comment 2:

Abstract-methods: January 2001 to inception: Do you mean January 2001 to present?
Please clearly describe your inclusion criteria based on PICO-S

Response 2:

We thank the reviewer for noting this error and have corrected it to 1st January 2001 to 29th October 2021. Due to character limitations in the abstract, we are unable to provide an extensive description in the abstract, but we have now added a clear description of the PICO-S to the eligibility criteria in the Methods section.

In summary, the population will consist of people with a post-viral syndrome (including Long COVID). We will include studies that have assessed the effectiveness of non-pharmacological interventions designed to improve symptoms of post-viral syndromes against standard care, an alternative non-pharmacological therapy, or a placebo. The outcomes will be changes in symptoms, exercise capacity, quality of life (including changes in mental health and wellbeing) and work capability. We will only include randomised controlled trials for studies on post-viral syndromes that predated the COVID-19 pandemic, but we will also include observational studies examining the emerging effectiveness of non-pharmacological interventions for Long COVID.

Comment 3:

Background:

Page 8, line 6: Please update the numbers and indicate the date as the cases are still evolving.

Page 8, paragraph 5: You may want to explain more about epidemiological features of post-viral syndromes to build a stronger support for your background.

Response 3:

We thank the reviewer and we have updated the numerical data on COVID-19 in response. Should the manuscript be deemed suitable for publication we will provide the latest data during the proof-reading phase. In response to the second point, we have now expanded on that paragraph providing an example of fatigue:

Like Long COVID, the signs and symptoms comprising these post-viral syndromes may include fatigue, musculoskeletal pain (e.g., joint pain), and mood (e.g. feeling depressed) and neurocognitive disturbances (e.g., difficulties with concentration and sleep).^{4,5} A particularly researched area has been the development of post-infectious fatigue following Epstein-Barr virus exposure.⁶ As we have demonstrated in our recently published review on the symptoms of Long COVID, fatigue is one of the most commonly reported symptoms.² Notably, over time interventions have been developed to support patients with Epstein-Barr post-infectious fatigue, such as the combination of cognitive-behavioural therapy with music therapy which has shown some promise.⁷ This example highlights the similarities that may exist between post-viral syndromes and the need to summarise evidence on their treatments, which may be relevant to those experiencing Long COVID.

Comment 4:

Objectives: Please indicate your PROSPERO registration number

Response 4:

We thank the reviewer for this point and have now added the PROSPERO registration number to that section.

“The review protocol has been registered with PROSPERO (CRD42021282074).”

Comment 5:

Methods:

Eligibility criteria-I see that this review intends to describe what is out there in terms of post-viral syndromes (symptoms, interventions, outcomes). However, for this review to be more transparent, I suggest providing a potential list of what you will consider included with regard to these variables (in a table or text). Especially this would be helpful for non-pharmacological treatments.

Outcomes and prioritisation: As mentioned above, providing a potential list of what this review will consider as outcomes in detail will help.

Response 5:

We thank the reviewer for highlighting this point. As we have now narrowed the focus of the review, we have removed the information detailing our search terms for symptoms and health impacts. Instead, we have enhanced our description of non-pharmacological treatments. However, for non-pharmacological treatments we have purposefully not listed all of the possible options in the search strategy as we did not want to limit the search to any type of intervention (such as graded exercise therapy, music therapy or pacing) in order to capture the full range of available treatments. Instead, we have now added some examples of suitable therapies to the main text:

The outcomes of the study will be changes in symptoms, exercise capacity, quality of life (including changes in mental health and wellbeing) and work capability. Although, these are numerous and heterogenous in nature, some examples of non-pharmacological treatments which may be used to support patients include pacing, cognitive behavioural therapy, and pulmonary rehabilitation. Additionally, where described in the included studies, we will also extract information on the symptoms and health impacts reported by participants in the included studies.

References

1. Ahmed H, Patel K, Greenwood DC, Halpin S, Lewthwaite P, Salawu A, et al. Long-term clinical outcomes in survivors of severe acute respiratory syndrome and Middle East respiratory syndrome coronavirus outbreaks after hospitalisation or ICU admission: A systematic review and meta-analysis. *J Rehabil Med* [Internet]. 2020 May 1 [cited 2021 Dec 20];52(5). Available from: <https://pubmed.ncbi.nlm.nih.gov/32449782/>
2. Aiyegbusi OL, Hughes SE, Turner G, Rivera SC, McMullan C, Chandan JS, et al. Symptoms, complications and management of long COVID: a review. *J R Soc Med* [Internet]. 2021 Sep 1 [cited 2021 Nov 28];114(9):428–42. Available from: <https://pubmed.ncbi.nlm.nih.gov/34265229/>
3. Staniszewska S, Brett J, Simera I, Seers K, Mockford C, Goodlad S, et al. GRIPP2 reporting checklists: Tools to improve reporting of patient and public involvement in research. *BMJ* [Internet]. 2017 Aug 2 [cited 2021 Mar 1];358:3453. Available from:

<http://dx.doi.org/10.1136/bmj.j3453><http://www.bmj.com/>

4. Hickie I, Davenport T, Wakefield D, Vollmer-Conna U, Cameron B, Vernon SD, et al. Post-infective and chronic fatigue syndromes precipitated by viral and non-viral pathogens: Prospective cohort study. *Br Med J* [Internet]. 2006 Sep 16 [cited 2021 Jul 4];333(7568):575–8. Available from: <http://www.bmj.com/>

5. Aucott JN, Rebman AW. Long-haul COVID: heed the lessons from other infection-triggered illnesses [Internet]. Vol. 397, *The Lancet*. Elsevier B.V.; 2021 [cited 2021 Jul 4]. p. 967–8. Available from: <https://doi.org/10.1101/2020.10.19.20214494>

6. White PD. What Causes Prolonged Fatigue after Infectious Mononucleosis—and Does It Tell Us Anything about Chronic Fatigue Syndrome? *J Infect Dis* [Internet]. 2007 Jul 1 [cited 2021 Nov 28];196(1):4–5. Available from: <https://academic.oup.com/jid/article/196/1/4/843612>

7. Malik S, Asprusten TT, Pedersen M, Mangersnes J, Trondalen G, Van Roy B, et al. Cognitive–behavioural therapy combined with music therapy for chronic fatigue following Epstein-Barr virus infection in adolescents: a randomised controlled trial. *BMJ Paediatr Open* [Internet]. 2020 Oct 1 [cited 2021 Nov 28];4(1):e000797. Available from: <https://bmjpaedsopen.bmj.com/content/4/1/e000797>

VERSION 2 – REVIEW

REVIEWER	Stephenson, Terence University College London, Institute of Child Health I know Kamlesh Khunti, Melanie Calvert and Shamil Haroon but they are not employed in my university. We have worked together on a review paper last year. This study type does not require ethical approval. Nevertheless, for the record I am the Chair of the HRA but do not participate personally in REC decisions.
REVIEW RETURNED	25-Jan-2022

GENERAL COMMENTS	1. Long COVID, long haulers; post-acute COVID syndrome are all terms used. The National Institute for Health and Care Excellence (NICE) definitions are: • Acute COVID-19: symptoms <4 weeks after confirmed infection• Ongoing symptomatic COVID-19: symptoms 4-12 weeks• Post-COVID-19 syndrome: >12 weeks The authors note that "A limitation of the review relates to the heterogeneity in the definition of post-viral syndromes as well as Long COVID, adding complexity to the synthesis of available evidence across the included syndromes". 'Long COVID', "post-COVID-19 condition" (WHO) and "post-COVID-19 condition" are all used. It might be worth these NIHR/UKRI funded researchers explaining that in the UK, NIHR/UKRI funded researchers continue to use long COVID because this is used by the public, healthcare professionals and in searches and systematic reviews. 2. Related to this, the authors should explain why they chose a 12 week cut-off "We will include studies with a population consisting of adults and/or children with post-viral syndromes persisting for more than 12 weeks" with reference to NICE and WHO definitions. This is touched on later with "However, similarly to the widely adopted definition of Long COVID" BUT which definition do they refer to which has been widely adopted? 3. Viruses identified in the last 3 years are excluded. Altho
--

	coronaviruses are not new, SARS-COV-2 has only been identified in the last 3 yrs. Is it included? This ambiguity needs clarifying. later they write "In summary, the population will consist of people with a post-viral syndrome (including Long COVID)" which seems at odds with "We will only include randomised controlled trials for studies on post-viral syndromes that predated the COVID-19 pandemic". 4. under 'Search methods for identification of studies', I couldn't see whether English language was an inclusion (nor could I see this in Suppl Info) or whether they would translate non-English articles?
--	--

REVIEWER	Khanpour Ardestani, Samaneh University of Alberta Faculty of Medicine and Dentistry, Pediatrics
REVIEW RETURNED	19-Jan-2022

GENERAL COMMENTS	I appreciate the authors great effort to clarify the objectives and methods of this valuable systematic review. Focusing on non-pharmacological treatments as the primary objective was a reasonable decision as the objectives in the previous version of the protocol were extensively broad. The methods section also is clearer now with PICO-S defined in the inclusion criteria. My comments have been addressed properly in this version and I suggest accepting the manuscript with some minor revisions mentioned below:  -Both in the introduction part of the abstract and the manuscript (suggesting last paragraph), you may briefly elaborate on why you have focused specifically on non-pharmacological treatments, so it corresponds better with your primary objective. -As you won't include the SRs anymore, you may remove the parts regarding using AMSTAR-2 for the risk of bias assessment. -Please re-check the manuscript for some minor typos and grammar errors.
--

VERSION 2 – AUTHOR RESPONSE

Reviewer 1:

Comment 1:

1. Long COVID, long haulers; post-acute COVID syndrome are all terms used. The National Institute for Health and Care Excellence (NICE) definitions are:

- Acute COVID-19: symptoms <4 weeks after confirmed infection
- Ongoing symptomatic COVID-19: symptoms 4-12 weeks
- Post-COVID-19 syndrome: >12 weeks

The authors note that "A limitation of the review relates to the heterogeneity in the definition of post-viral syndromes as well as Long COVID, adding complexity to the synthesis of available evidence across the included syndromes". 'Long COVID', "post-COVID-19 condition" (WHO) and "post-COVID-19 condition" are all used. It might be worth these NIHR/UKRI funded researchers explaining that in the UK, NIHR/UKRI funded researchers continue to use long COVID because this is used by the public, healthcare professionals and in searches and systematic reviews.

Response 1:

We thank Prof Stephenson for re-reviewing the protocol and providing further helpful comments to improve the manuscript.

We have now added the following paragraph to the introduction to highlight this point:

There is considerable heterogeneity in the clinical definition of ongoing symptoms following initial

exposure to SARS CoV-2. The National Institute for Health and Care Excellence (NICE) define symptoms persisting beyond 4 weeks as either 'ongoing symptomatic COVID-19' (symptoms lasting between 4-12 weeks) or 'Post covid-19 syndrome' (symptoms lasting beyond 12 weeks).¹ Globally other terms such as 'post-COVID condition' or 'long haulers' are used. However, in the UK the more commonly adopted and preferred term among healthcare professionals and patient groups for ongoing symptoms following initial viral exposure to SARS CoV-2 is 'Long COVID,' the term adopted for this review.

Comment 2:

2. Related to this, the authors should explain why they chose a 12 week cut-off "We will include studies with a population consisting of adults and/or children with post-viral syndromes persisting for more than 12 weeks" with reference to NICE and WHO definitions. This is touched on later with "However, similarly to the widely adopted definition of Long COVID" BUT which definition do they refer to which has been widely adopted?

Response 2:

We thank the reviewer for this comment. We apologise as we had originally described this criterion with different levels of precision in the protocol. As there is no confirmed definition of post viral syndromes often with many research studies describing their temporal consequences with little mention of timeframe, we have kept to a 12-week criterion to define persistent post-viral effects to be consistent with the definition of Long COVID now commonly requiring symptoms to be present for at least 12 weeks.² We have explained this in the manuscript as follows:

We will include studies with a population consisting of adults and/or children with post-viral syndromes.

To the authors' knowledge there is no internationally agreed definition of post-viral syndromes and as such there is likely to be heterogeneity in the temporal description of syndrome onset following the initial viral exposure. Where temporal criteria are included, we will aim to include those with symptoms lasting beyond 12 weeks (in line with the minimum timeframe for Long COVID used by the World Health Organization (WHO)).² However we will also include publications which provide no firm timeframe but indicate an aspect of chronicity or symptoms persisting for a prolonged period.

Comment 3:

3. Viruses identified in the last 3 years are excluded. Altho coronaviruses are not new, SARS-COV-2 has only been identified in the last 3 yrs. Is it included? This ambiguity needs clarifying. later they write "In summary, the population will consist of people with a post-viral syndrome (including Long COVID)" which seems at odds with "We will only include randomised controlled trials for studies on post-viral syndromes that predated the COVID-19 pandemic".

Response 3:

We thank the reviewer for identifying this and can confirm that we have reworded the according sections to clarify this ambiguity.

Predominantly asymptomatic viruses, viruses identified in the last 3 years (except for SARS CoV-2), and viruses predominantly existing as zoonoses will all be excluded, in addition to studies on chronic viral infections including Hepatitis B, Hepatitis C and HIV post-viral syndromes.

In summary, the population will consist of either people with a post-viral syndrome who contributed to a randomised controlled trial or those with Long COVID who either contributed to an observational study or a randomised controlled trial. We will include studies that have assessed the effectiveness of non-pharmacological interventions designed to improve symptoms of post-viral syndromes against standard care, an alternative non-pharmacological therapy, or a placebo. The outcomes will be changes in symptoms, exercise capacity, quality of life (including changes in mental health and wellbeing), and work capability.

Comment 4:

4. under 'Search methods for identification of studies', I couldn't see whether English language was an

inclusion (nor could I see this in Suppl Info) or whether they would translate non-English articles?

Response 4:

We thank the reviewer for asking us to clarify this. We have updated the 'Search methods for identification of studies' to clarify our inclusion criteria.

Studies will not be excluded on language. Where studies are not published in English then they will be translated using Google Translate or a by a member of staff at the Institute of Applied Health Research.

Reviewer 2:

Comment 1:

I appreciate the authors great effort to clarify the objectives and methods of this valuable systematic review. Focusing on non-pharmacological treatments as the primary objective was a reasonable decision as the objectives in the previous version of the protocol were extensively broad. The methods section also is clearer now with PICO-S defined in the inclusion criteria. My comments have been addressed properly in this version and I suggest accepting the manuscript with some minor revisions mentioned below.

Response 1:

We thank Dr Ardestani for her kind comments about the review, and the way that we addressed her comments. We are glad that the objectives and methods is clearer, and we addressed her comments fully.

Comment 2:

Both in the introduction part of the abstract and the manuscript (suggesting last paragraph), you may briefly elaborate on why you have focused specifically on non-pharmacological treatments, so it corresponds better with your primary objective.

Response 2:

We thank the reviewer for this comment and have added a sentence at the end of the abstract introduction and at the end of the last paragraph of the introduction in the manuscript.

To minimise overlap with existing work in the field exploring pharmacological treatments for Long COVID and on the basis of the advice of our expert advisory group, we have chosen to focus on non-pharmacological treatments.

Comment 3:

As you won't include the SRs anymore, you may remove the parts regarding using AMSTAR-2 for the risk of bias assessment.

Response 3:

We thank the reviewer for highlighting this, as it is no longer suitable. We have now removed the parts regarding the use of AMSTAR-2 for the risk of bias assessment from the abstract, and methods section.

Comment 4:

Please re-check the manuscript for some minor typos and grammar errors.

Comment 4:

We thank the reviewer for highlighting to us some minor type and grammar errors. It has been proofread and corrections made as appropriate.

References:

1 National Institute of Health and Care Excellence. Identifying people with ongoing symptomatic COVID-19 or post-COVID-19 syndrome | COVID-19 rapid guideline: managing the long-term effects of COVID-19. 2020.<https://www.nice.org.uk/guidance/ng188/chapter/1-Identifying-people-with-ongoing-symptomatic-COVID-19-or-post-COVID-19-syndrome> (accessed 10 Sep2021).

2 World Health Organization. A clinical case definition of post COVID-19 condition by a Delphi consensus, 6 October 2021. 2021.https://www.who.int/publications/i/item/WHO-2019-nCoV-Post_COVID-19_condition-Clinical_case_definition-2021.1 (accessed 2 Mar2022).